# Automated systematic evaluation of cryo-EM specimens with SmartScope

**Jonathan Bouvette[1†], Qinwen Huang[2†], Amanda A Riccio[1], William C Copeland[1], Alberto Bartesaghi[2,3,4]\*, Mario J Borgnia[1]\***

[1]Genome Integrity and Structural Biology Laboratory, National Institute of Environmental Health Sciences, Research Triangle Park, United States; [2]Department of Computer Science, Duke University, Durham, United States; [3]Department of Electrical and Computer Engineering, Duke University, Durham, United States; [4]Department of Biochemistry, Duke University School of Medicine, Durham, United States

**Abstract** Finding the conditions to stabilize a macromolecular target for imaging remains the most critical barrier to determining its structure by cryo-electron microscopy (cryo-EM). While automation has significantly increased the speed of data collection, specimens are still screened manually, a laborious and subjective task that often determines the success of a project. Here, we present SmartScope, the first framework to streamline, standardize, and automate specimen evaluation in cryo-EM. SmartScope employs deep-learning-based object detection to identify and classify features suitable for imaging, allowing it to perform thorough specimen screening in a fully automated manner. A web interface provides remote control over the automated operation of the microscope in real time and access to images and annotation tools. Manual annotations can be used to re-train the feature recognition models, leading to improvements in performance. Our automated tool for systematic evaluation of specimens streamlines structure determination and lowers the barrier of adoption for cryo-EM.

**\*For correspondence:**
alberto.bartesaghi@duke.edu (AB);
mborgnia@nih.gov (MJB)

[†]These authors contributed equally to this work

**Competing interest:** The authors declare that no competing interests exist.

## Editor's evaluation

This paper describes a new software tool: SmartScope, for automated screening of cryo-EM grids. SmartScope can also perform automated data collection on suitable grids, including with beam-image shifts and tilted stage geometries. If it works in practice as advertised in the paper, then it will be a highly useful tool for the field, especially if other groups would also contribute to its open-source and modular code.

## Introduction

Over the past decade, advances in hardware and software have improved the resolution and throughput of single particle analysis (SPA), establishing cryo-EM as a method of choice in structural biology. However, optimizing specimens for high-resolution cryo-EM imaging remains a significant barrier (*Weissenberger et al., 2021*). The ideal specimen for solving a structure is a single layer of randomly oriented macromolecular complexes embedded into a thin slab of vitreous ice. During specimen preparation, interactions with the air-water interface facilitated by the confinement into a thin layer of buffer can destabilize protein complexes leading to denaturation and aggregation or force the molecules into a 'preferred orientation' (*Noble et al., 2018*). In addition, vitrification methods typically yield variations in ice thickness across the grid. These artifacts can severely limit the quality of specimens and are typically addressed through an optimization process in which several

parameters are varied to increase the stability and mono-dispersity of the target macromolecule (*Passmore and Russo, 2016*). Evaluating each combination of parameters involves comprehensive sampling of one or more grids using a cryo-EM. Testing all combinations is impractical because the number grows significantly with the inclusion of each parameter. Instead, an iterative search is performed in which a limited number of parameters are evaluated, and new conditions are selected based on the results.

The goal of specimen screening is to learn as much as possible from each condition, often taking advantage of the heterogeneous landscape of each grid to extract valuable information about the behavior of the macromolecule of interest. This process involves selecting areas for evaluation, adjusting the positioning and optical conditions of the microscope, and recording images at multiple magnifications. The lowest magnifications are used to assess the overall quality of the vitrification process, the number of potential areas amenable to sampling at higher resolution and some macroscopic indicators of sample instability such as aggregation. Higher magnification images provide direct information about the macromolecules of interest, such as particle integrity, distribution, affinity for the substrate, density, heterogeneity and orientation, as well as the quality of the ice, and the resolution limit of the images. This makes manual specimen screening a time-consuming activity with a steep learning curve in which the implicit subjectivity in the selection of areas can lead to suboptimal sampling, resulting in missing information. The quality of the results, the speed, and even the integrity of the instrument, all depend on the experience and skills of the operator.

Existing software for automated cryo-EM is not designed to provide a thorough sampling of each grid. Instead, packages are optimized for acquiring a large number of high-quality images of a preselected set of targets (*Mastronarde, 2005*; *Suloway et al., 2005*). Although all data collection packages preserve lower magnification images and the associated stage positions, they are not designed to facilitate virtual navigation of the grid with the exception Leginon's companion software, Appion (*Lander et al., 2009*), which provides offline access to the results via a web user interface (WebUI). However, none of the existing packages is optimized for screening, nor they provide a web-based solution for controlling the microscope during specimen evaluation.

Here, we present SmartScope, a web-based, highly available expert system capable of performing unsupervised screening of specimens and automated data collection for cryo-EM. SmartScope uses pretrained generalized deep learning (DL) models for feature detection and selection to maximize sampling and provide information to guide specimen optimization. By combining automation, machine learning, and remote control, we aim to increase the efficiency of the screening process, reducing costs and dramatically increasing availability. Finally, SmartScope is designed as a modular framework, facilitating the addition of new algorithms for area selection and navigation that can further improve targeting performance.

## Results

The complexity of a screening workflow depends on several factors including the instrument used and the type of specimen. Here, we describe the extended operation of a microscope furnished with an autoloader device and loaded with frozen hydrated targets for SPA, which are prepared on a micro-patterned holey substrate or continuous carbon (*Figure 1* and *Figure 1—figure supplement 1*). All the interactions with the software are carried out using the SmartScope WebUI which also allows to monitor progress and control the workflow with little to no training in cryo-EM. Further, SmartScope permits simultaneous access from multiple remote devices, greatly facilitating collaborative work.

### Initialization

After a cassette is inserted in the autoloader, a session is initialized by providing the list of grids to be evaluated along with a series of parameters applicable to all of them (*Appendix 1—table 1*). SmartScope then initiates a connection to SerialEM (*Mastronarde, 2005*) via its python API to issue commands to the microscope. This connection is locked to prevent the simultaneous execution of multiple workflows. For instruments equipped with automated loading systems, grids are loaded sequentially into the column and subjected to the operations described in the sections below.

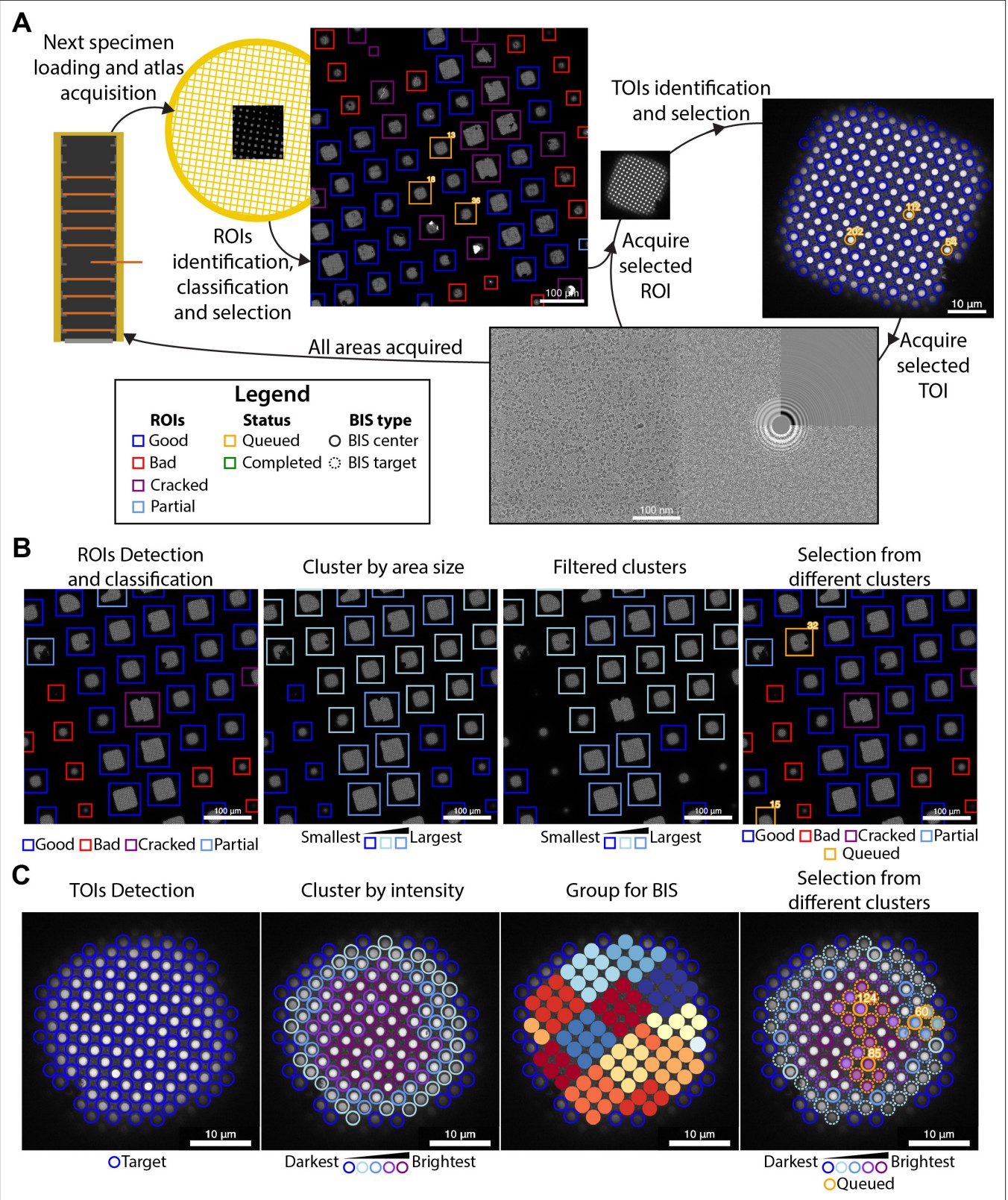

**Figure 1.** Overview of the SmartScope framework. (**A**) Workflow for unsupervised grid navigation and imaging. SmartScope handles specimen exchange, atlas acquisition, regions of interest (ROIs) identification, classification, and selection. It then visits the selected regions and identifies and selects targets of interest (TOIs) which are acquired at higher magnification and preprocessed. (**B**) Detailed steps in ROI selection. After detection and classification, ROIs are also clustered into groups. In the example is a clustering by size. Then, from the ROIs are queried based on their class and ROIs

*Figure 1 continued on next page*

*Figure 1 continued*

from different clusters are selected. (**C**) Detailed steps in TOI selection. Shown here is the hole detection followed by a median intensity clustering. Then, holes are grouped by image-shift radius and groups from each cluster are selected for imaging.

The online version of this article includes the following figure supplement(s) for figure 1:

**Figure supplement 1.** Detailed SmartScope workflow.

**Figure supplement 2.** Beam-image shift hole grouping algorithm.

**Figure supplement 3.** Overall software architecture of SmartScope.

## Grid analysis

For each grid loaded, a series of low magnification images are acquired and stitched together to generate a grid map or 'Atlas' which is analyzed by SmartScope's DL driven window detector and classifier (*Figure 1A*). Windows deemed suboptimal for imaging due to physical damage or heavy contamination are excluded from further analysis. The remaining 'good' windows are reclassified and clustered based on a selectable criterion (e.g. areas suitable for imaging). Representatives of each window cluster are added to a list of regions of interest (ROIs) with the goal of adequately sampling the diversity of imageable areas (*Figure 1B*). The program then proceeds to visit and select imaging targets from the ROIs in this list, which can be modified via the WebUI at any time before the grid is completed.

## Selection of targets

The stage is moved to the next ROI, brought to eucentric height, and imaged at a magnification that ensures complete coverage of the area (*Figure 1A*). The next step is to identify targets of interest (TOIs) based on a programmable criterion that depends on the specimen. For example, SmartScope's DL based hole finder is used to detect holes in frozen hydrated SPA specimens. The current algorithm classifies the holes based on their average signal intensity (a proxy for ice thickness) and clusters them into a selectable number of groups. By default, the group containing the darkest targets is rejected as not suitable for imaging. To maximize diversity, holes are selected from the different clusters and added to a TOI list (*Figure 1C*). As with ROIs, this selection can be modified at any time during imaging of the grid. A protocol for selecting negative stain TOIs is also available and plug-ins for other types of specimens may be incorporated in the future.

Selected TOIs are visited sequentially by moving the stage to their predicted coordinates. A series of images, at a magnification that encompasses the TOI and surrounding area, are used to recenter the imaging area on the target (a hole in the substrate for SPA). These intermediate resolution images are stored in the database as they often provide valuable information about the specimen, such as affinity of the macromolecules for the support material, aggregation, denaturation, etc. Autofocus and drift stabilization procedures are then performed before acquiring high-magnification images of the target. An optional random offset from the center of each hole can be specified to capture images at different distances from the edge of the substrate. The newly acquired images are processed using the routine alignframes in IMOD (*Kremer et al., 1996*) and the program CTFFIND4 (*Rohou and Grigorieff, 2015*) to facilitate assessment of data quality. This cycle is repeated until all TOIs are imaged; then the workflow proceeds to the next ROI.

After all selected ROIs for the grid is finished, SmartScope automatically switches to the next grid. This behavior may be modified by selecting the 'pause between grids' option on the session menu. Pausing allows for the selection of additional ROIs at the end of the cycle, providing better control of unattended sessions or when evaluating unusual specimens where automated sampling may not be satisfactory.

## Accessing and annotating results

SmartScope systematically documents the results and facilitates their analysis. During collection, all images and their related metadata are stored in a consistent data structure. To display and interact with these data, SmartScope implements an intuitive WebUI that tracks the imaging process in real time. Moreover, it enables remote interaction with a running session, such as modifying area selection, changing labels and acquisition parameters, and taking notes about the specimen, all without interrupting the acquisition workflow.

After a session is over, SmartScope can automatically copy the data to long-term or object storage. The data remains available through the WebUI and allows users to make additional annotations. Other tools, such as micrograph curation and exporting of metadata as star files are also available.

## Tools for exhaustive screening and high-throughput data collection

SmartScope can also perform high-throughput data collection. A session can be initialized in data collection mode or changed from screening to data collection by setting the number of TOI to sample to zero. This will select all the available TOIs for imaging.

To achieve high-throughput, SmartScope makes uses beam-image shift (BIS) for multi-hole imaging (*Cheng et al., 2018*). BIS can be used during screening for more exhaustive sampling, allowing for exploratory data collections that can provide enough images to carry out 2D classification or initial 3D reconstruction. The BIS grouping in SmartScope uses an algorithm that groups the holes within a given radius (*Figure 1—figure supplement 2*). Only the targets that are labeled as 'good' and are in the included clusters are used for the grouping. The algorithm attempts to maximize the coverage of targets while minimizing the total number of groups. To maximize the speed of data collection, a minimal group size can also be specified to prevent the algorithm from assigning small groups of holes. The BIS radius and minimal group size is specified at the start of each session and can be modified during the session.

One way to alleviate the commonly occurring problem of orientation bias in single-particle cryo-EM is to collect data on a tilted specimen (*Tan et al., 2017*). SmartScope can perform BIS data collection on tilted specimens, where the position and defocus of the targets are corrected using geometrical tilt constraints with a throughput that is similar to regular non-tilted data collection. Moreover, the tilt angle can be seamlessly changed at any point during the acquisition process. Combined with the integrated in-line data processing, this allows to adapt the data collection strategy on-the-fly based on the newly acquired knowledge.

The combination of automated operations such as feature detection routines, multi-hole imaging, and tilted data collection capabilities, makes SmartScope a powerful tool that can accelerate screening of cryo-EM samples and achieve high-throughput data collection.

## Asynchronous imaging and processing

SmartScope was designed to maximize microscope efficiency and to remove much of the idling time in the imaging process. A common source of idling is the processing time required for rendering the frame averages or calculating the CTF fits. To minimize the impact of this, the microscope's imaging process and the image processing routines run as parallel processes. The newly acquired images or movies are queued up for analysis and processed sequentially on a separate thread. For the atlas and windows, processing includes detecting, classifying, and selecting targets. For high-magnification TOIs, it involves frame alignment when fractions are saved, and CTF estimation. This allows the microscope to immediately acquire the next target while the images are being analyzed (*Figure 1—figure supplement 1*).

## Installation and configuration

SmartScope bundles a web server for the WebUI, a database server and the core package necessary to run the main imaging workflow. It relies on the database to store and query essential metadata (*Figure 1—figure supplement 3*). To simplify the deployment and orchestration of these services, we created a Docker image that should be compatible with most Linux systems.

The application can be installed on a single workstation that will handle the execution of all the services. It can also be installed in a master-worker configuration where, for example, one computer handles the web server and the database (master), while the main workflow is executed on a workstation that is connected to the microscope network (worker). The minimal requirements are that all the systems can access the database that holds the metadata for the session and targets, and the filesystem where the images are saved.

A long-term storage area that holds the data from previous sessions can be specified. Both mounted network drives and object-stores can be used to store the data. This allows to clear data from the main local drives leaving space for the ongoing sessions while keeping older data accessible through the WebUI.

After installation, administrators can login to the WebUI management portal where microscope and detector information needs to be added to allow connection to the instruments. Multiple microscopes can be installed on a single instance of SmartScope, serving as a central hub for microscope access. To access the server, each user has an account and groups are created by an administrator. Users can only access the data from the groups they belong to.

In SerialEM, a settings file containing the magnification for low-dose imaging needs to be prepared for each microscope and detector. The conditions should be set as follows: full square image bound to the Search preset using a magnification that shows the entire square; the fine hole re-alignment condition in low SA magnification bound to the View preset; the data acquisition conditions bound to the Record preset with the acquisition and dose fractionation settings bound to the Preview. As different hardware combinations would require different settings, we included a table with the settings used on a Talos Arctica equipped with a K2 detector and Titan Krios G4 equipped a K3 detector and biocontinuum energy filter as guidelines (*Appendix 1—table 2*).

## Automated object detection and classification

SmartScope identifies and classifies ROIs and TOIs suitable for cryo-EM imaging using DL approaches. At the atlas level, areas suitable for imaging appear as 'windows,' commonly shaped as squares, through the metallic grid in which the support layer is intact and not blocked by thick ice or large contaminants. Windows are automatically detected and classified using a pretrained Region-based Convolutional Neural Network (*Girshick, 2015*) that identifies the 'good' windows with 80% precision (*Figure 2A*, *Figure 2—figure supplement 1A*, *Appendix 1—table 3*), thus providing information that can effectively guide the instrument to avoid undesirable regions of the grid.

Selected windows are then acquired at a higher magnification where TOIs can take various shapes depending on whether the modality is single particle cryo-EM, tomography, or negative stain. In single particle cryo-EM, these TOIs usually show as holes in the substrate and are difficult to detect with traditional image processing tools due to the low contrast, especially when carbon mesh grids are used. We implemented a robust hole detector for frozen specimens based on the You-Only-Look-Once (*Redmon et al., 2016*) object detection architecture (*Figure 2B* and *Figure 2—figure supplement 1B*). To prevent the network from incorrectly picking dark ice contaminants, we also added a classification step to separate holes from contaminants and were able to correctly identify 89% of the holes.

## Screening mode statistics

In screening mode, the average time required for exchanging specimens and acquiring a partial atlas covering at least 25% of the grid surface is 7.8 min (*Figure 3*, *Appendix 1—table 4* and *Appendix 1—table 5*). The median sampling time for a specimen is 21 min, yielding a median of 9.0 high-magnification images of holes sampled from 3.0 different windows. Each day, our screening microscope thoroughly screens an average of 16 specimens and performs data collection for approximately 16 hr.

## Data collection mode statistics

SmartScope offers a convenient option to set up, track data collection and to label and annotate exposures. In data collection mode, the microscope continuously acquires areas and finds targets using operator assistance only to fine tune the selection to specific needs, significantly reducing setup time as compared to our manual workflow. The median data collection setup time, from specimen loading to the start of high-magnification acquisition, is 32 min with our K2 detector (*Appendix 1—table 6*). As an example, we used SmartScope to determine a 3.4 Å map of the 55 kDa homodimer accessory subunit of the human mitochondrial DNA polymerase (*Young and Copeland, 2013*) (EMD-25764, *Figure 4*, *Figure 4—figure supplement 1*, *Figure 4—figure supplement 2*). For this specimen, we collected 4327 micrographs and seamlessly tilted to 30° for the last third of the dataset to improve the angular sampling of the protein (*Table 1*).

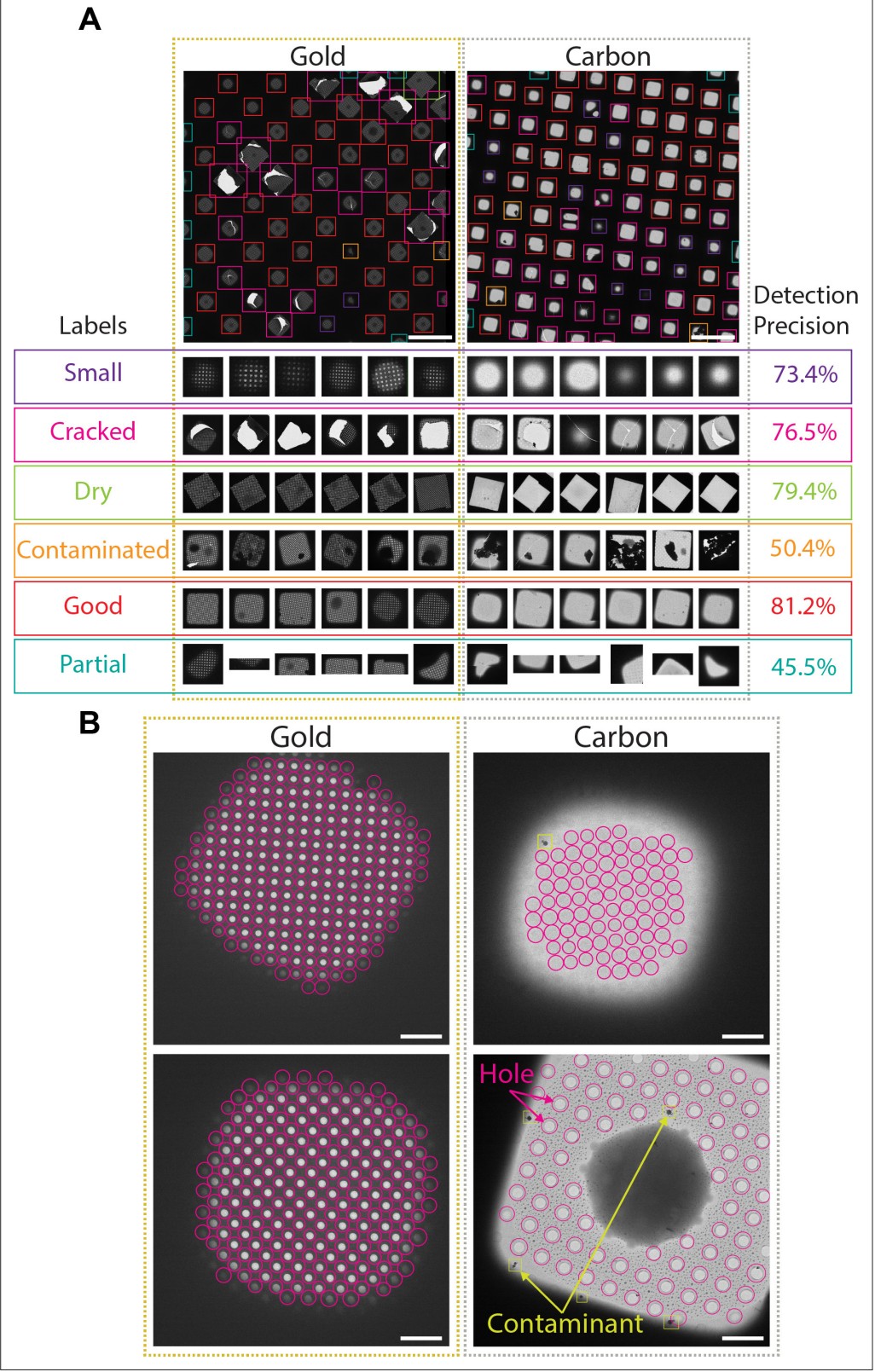

**Figure 2.** Deep-learning-based feature recognition for autonomous grid navigation. Sample images show the performance of the square (**A**) and hole (**B**) detectors applied to gold (left) and carbon (right) grids. (**A**) Automatic detection of squares and classification into six different classes: small, cracked, dry, contaminated, good, and partial (white scale bars are 100 μm). Representative examples of squares assigned to each class

*Figure 2 continued on next page*

*Figure 2 continued*

and corresponding detection precision values are shown (bottom panel). (**B**) Hole detection performance on representative square images extracted from gold and carbon grids. The hole detector implements a classification step to filter out contaminants (shown in yellow) and increases hole detection precision (shown as pink circles) (white scale bars are 10 µm).

The online version of this article includes the following figure supplement(s) for figure 2:

**Figure supplement 1.** Object detection training strategy.

## Discussion

SmartScope is the first package specifically designed to assist, document, and automate specimen evaluation during the process of optimizing samples for cryo-EM. The program delivers a unique user experience through a WebUI that provides live remote supervision and control of the screening process using a standard web browser. The same WebUI facilitates analysis of results at any time during and after a session. Multiple users can access the same live or stored session simultaneously and multiple instruments can be controlled from the same server. Automated navigation routines provide control of the microscope without granting full access to functions that may compromise the integrity of the instrument. SmartScope uses fast and robust AI-driven feature recognition algorithms to fully automate the cryo-EM imaging workflow. The steps of target identification, object classification, and clustering offer a powerful way to sample a wide variety of areas during screening and help determine the next steps in specimen optimization. This enables complete unsupervised execution of a screening workflow as well as supervised exploration with minimal user training.

SmartScope can also collect data in a semi-supervised or fully automated way. The areas automatically selected by SmartScope can be modified interactively or programmatically without interrupting the process of data collection. This maximizes the use of the microscope and offers the possibility of integrating feedback from in-line data processing workflows to adaptively improve image quality during acquisition, without user intervention. Unsupervised multi-specimen screening and short exploratory data collection sessions can be scheduled to run overnight, offering new ways of using the microscope.

The interface to the microscope hardware in SmartScope is currently achieved through SerialEM, which provides abstraction interfaces to the main microscope and detector manufacturers as well as being open source and well supported. However, the current integration is made so that interfaces to other software (e.g. Leginon, Digital Micrograph, SmartEPU) can also be integrated with SmartScope in the future.

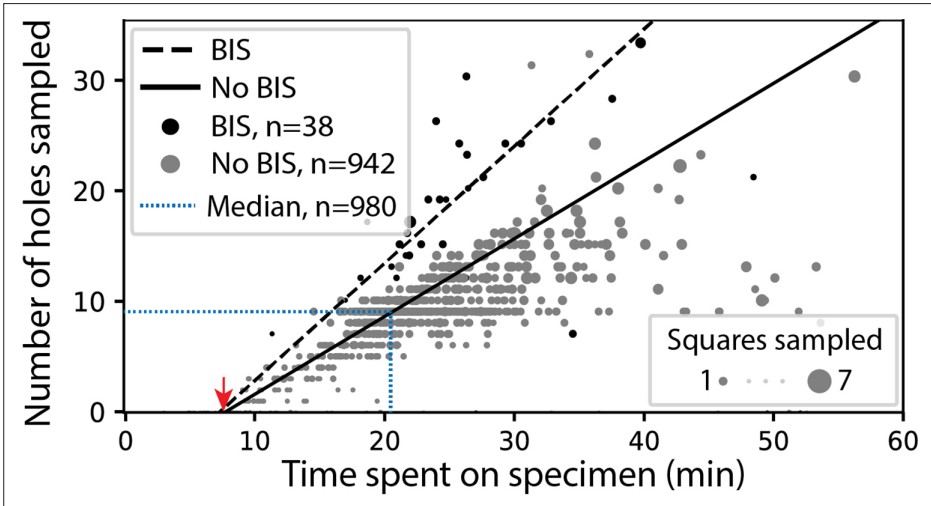

**Figure 3.** SmartScope's screening mode statistics. Screening rates with and without beam-image shift (BIS) were 1.0 and 0.7 holes per minute, respectively (RANSAC regression). The red arrow indicates the time of specimen loading and start of atlas acquisition. Dashed blue line represents the median session duration (21.6min) and the median number of high-magnification images (9.0) obtained per specimen during screening mode.

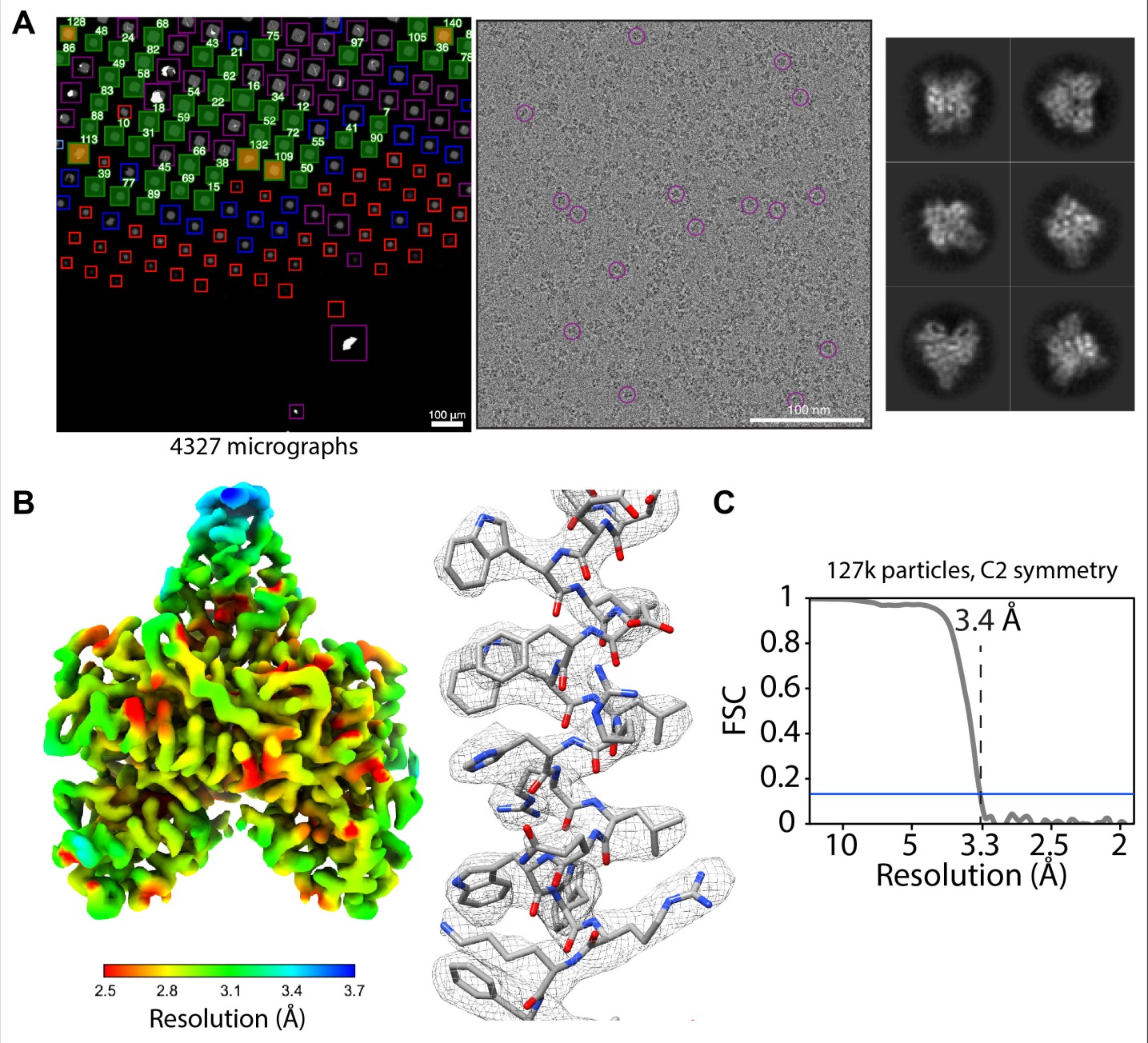

**Figure 4.** Acquisition of POLG2 dataset using SmartScope. (**A**) Atlas of the specimen (left), typical micrograph (center) with some particles picked (purple circles) and 2D classes of POLG2 (right). (**B**) Resulting map of POLG2 colored by local resolution (left) and example of an alpha helix with atomic model fit into the density (left). (**C**) Masked Fourier-shell correlation curve between half-maps showing a resolution of 3.4Å.

The online version of this article includes the following figure supplement(s) for figure 4:

**Figure supplement 1.** Structure determination of POLG2 using SmartScope.

**Figure supplement 2.** POLG2 particles from the areas collected at 0° tilt, 30° tilt, and combined.

SmartScope has a modular design where new object detection and classification algorithms can be added as plugins, allowing integration of existing object detection, and area selection programs for cryo-EM (*Fan et al., 2022*; *Kim et al., 2021*; *Rheinberger et al., 2021*; *Schorb et al., 2019*; *Xu et al., 2020*; *Yokoyama et al., 2020*; *Yonekura et al., 2021*). Additionally, the ability to use multiple feature detection and clustering methods at different magnification levels enables the creation of customized protocols for specific applications. This provides flexibility to optimize the selection of areas on

**Table 1.** Cryo-EM data acquisition parameters and statistics.

| | POLG2 (EMD-25764) | | |
| --- | --- | --- | --- |
| | **No tilt** | **Tilted** | **Combined** |
| **Hardware** | | | |
| Microscope | Talos Arctica (Thermo Fisher) | | |
| Detector | K2 summit (Gatan Inc) | | |
| **Data collection and processing** | | | |
| Magnification | 45,000 | | |
| Voltage (kV) | 200 | | |
| Electron exposure (e⁻/Å²) | 50 | | |
| Defocus range (μm) | 1.2–1.8 | 1.4–1.6 | 1.2–1.8 |
| Tilt angle (°) | 0 | 30 | |
| Pixel size (Å/pixel) | 0.932 | | |
| Movie No. | 3029 (70%) | 1282 (30%) | 4311 |
| Symmetry imposed | C2 | | |
| Final particles | 99,189 (78%) | 28,641 (22%) | 127,860 |
| Map resolution (unmasked) | 3.7 | 4.3 | 3.7 |
| Map resolution (masked) | 3.5 | 3.9 | 3.4 |
| FSC threshold | 0.143 | | |
| **Data collection statistics** | | | |
| Setup time* (min) | 60 | | |
| Throughput (movies/hr) | 117.9 | | |
| Throughput last hour (movies) | 130 | | |

*Data collection times are calculated as the time needed from grid loading to the collection of the first 50 high-magnification targets minus the time required for these 50 targets to be acquired.

a wider variety of targets in cryo-EM, such as virions, filaments, and cells. Finally, as we gather more data about difficult specimens and edge cases, we envision the establishment of a globally accessible 'virtual microscopist' server capable of improving itself through periodic re-training based on voluntarily submitted labeled datasets.

SmartScope has proven to be an extremely valuable tool in our facility. It has streamlined book-keeping, which in turn resulted in better decision making for specimen optimization. It has also maximized microscope usage by eliminating idling time, reducing setup, and screening times. SmartScope facilitates data and instrumentation access as well as collaboration by easing access to cryo-EM technologies and improving the way cryo-EM experiments are carried out. With specimen screening as a primary focus, SmartScope addresses an important limiting step in cryo-EM.

# Materials and methods
## Cryo-EM
All the data presented in this study was acquired on a Talos Arctica (Thermo Fisher Scientific) operating at 200 kV and equipped with a K2 direct-electron detector (Gatan Inc). SmartScope was also tested on a Ceta CMOS detector and a Titan Krios (Thermo Fisher Scientific) equipped with a K3 detector and BioQuantum energy filter (Gatan Inc). The statistics were derived from data acquired exclusively with the K2 detector. For microscope and detector control, SmartScope uses SerialEM 4.0 through the python API library (*Mastronarde, 2005*).

## Square finder

To localize and classify square windows, a Faster R-CNN-based framework (*Girshick, 2015*) that uses a ResNet50 architecture as the feature extraction backbone was adopted. It incorporates a feature pyramid network for identification of objects at different magnification levels. In addition, since most features have approximately equal width and height, the bounding boxes were constrained to have aspect ratios within the 0.8–1.2 range. To improve robustness and stability of the model, data augmentation was applied to the training data, including zoom-in/zoom-out, rotation, contrast adjustments, and flipping. The degree of augmentation for the contrast intensity was limited to the 0.8 and 1.2 range. To compensate for label imbalance, random oversampling was added during training. Squares are classified into six different classes: good (suitable for imaging), small (thick ice), contaminated, cracked, fractioned, and broken. The low-level magnification feature detector was trained using a total of 26 atlases from both carbon and gold mesh grids acquired on Ceta (Thermo Fisher Scientific) and K2 (Gatan Inc) detectors. Each atlas contains around 50–100 squares on average. The original atlases, usually having widths and heights greater than 10,000 pixels were downsampled to 2048 × 2048 pixels to reduce memory requirements. The framework is implemented using the python library Detectron2 (*Wu et al., 2019*). Training the detector takes around 2 hr when running on a NVIDIA TITAN V GPU card with 32 GB of RAM. The pre-trained weights are then used for fast real-time square detection during screening, which can evaluate each atlas image in under a second.

## Hole finder

To identify holes in all grid types and contrast levels, a deep neural-network architecture based on the YOLOv5 model was adopted (*Jocher et al., 2020*; *Redmon et al., 2016*). We used the Cross Stage Partial Network (CSPNet) (*Wang et al., 2019*) as the feature extraction backbone and standard convolutional layers as detection layers. Since holes have circular shapes, the aspect ratios of the bounding boxes were also constrained. To further facilitate training, instead of using arbitrary numbers for anchor bounding boxes generation, clustering algorithms to the ground truth boxes from the training dataset were applied to find the most common occurring sizes and we used these sizes to determine the anchor bounding box sizes. Data augmentation was applied during training, including contrast/brightness adjustment, rotation/translation, zoom-in/zoom-out, and cropping. To deal with small contamination areas that can be incorrectly detected as holes, an additional 'contaminants' class is used to filter out such areas. Training of the hole finder was done using 36 square images acquired on Ceta (Thermo Fisher Scientic) and K2 (Gatan Inc) detectors and took 1.5 hr when running on a NVIDIA TITAN V GPU card with 32 GB of RAM and inference takes less than a second. For memory efficiency, each square was resized to 1280 × 1280 pixels.

## POLG2 purification

Protein was expressed and purified essentially as described (*Young et al., 2015*) with the following exception: Triton-X was removed from all steps following lysis. Following Ni purification, pooled protein containing His$_6$-POLG2, as determined by SDS-PAGE, was injected onto a monoS column. Protein was eluted from S column in a linear gradient from 5 to 50% Buffer B (25 mM HEPES, 1 M NaCl, 10% glycerol, 1 mM EDTA, 1 mM TCEP). Peak fractions were eluted around 310 mM NaCl. Fractions were checked for purity, combined, and concentrated using an Amicon concentrator (Millipore) to 28 µM. Protein is flash frozen and stored at –80 °C.

4.1 µM (monomer) his-tag POLG2 was incubated in a 1:1 molar ratio with FORK1 DNA as previously described (6) Oligonucleotides: (i) DCRANDOM- 44 ACTTGAATGCGGCTTAGTATGCATTGTA AAACGACGGCCAGTGC (2) TSTEM GCACTGGCCGTCGTTTTACGGTCGTGACTGGGAAAACCCT GGCG (3) U25 CGCCAGGGTTTTCCCAGTCACGACC were all purchased from IDT. Protein in a final buffer of 20 mM HEPES pH 8, 1.5 mM Tris pH 7.5, 30 mM KCl, 50 mM imidazole pH 8, 0.3 mM EDTA, 225 mM NaCl, 1.5% glycerol, 1 mM TCEP was incubated on ice for approximately 30 min before grid application.

## Cryo-EM specimen preparation

UltraAUfoil R1.2/3 (Quantifoil Micro Tools GmbH) grids were glow-discharged on both sides for 30 s at 15 mA using a Pelco Easiglow. 3 µL of the final buffer was deposited on the back of the grid and 3 µL of POLG2 sample was deposited on front side of the grid. Excess sample was blotted 4 s with

blotting force –1, the chamber set at 12 °C and 95% humidity using a Vitrobot Mark IV (Thermo Fisher Scientific).

## Data collection of POLG2 with SmartScope

Data was collected on a Talos Arctica (Thermo Fisher Scientific) operating at 200 kV equipped with a Gatan K2 direct electron detector (Gatan Inc). Data collection was set up using SmartScope using a 6 × 6 tile atlas, image-shift grouping radius of 4 μm and minimum group size of four holes, rolling target defocus of –1.2 to –1.8 μm and drift settling threshold at 1 Å/s. A total of 4311 60-frame movies were collected at a 0.932 Å/pixel and a total dose of 54 e⁻/Å². 3029 movies were collected at 0° tilt angle and 1282 movies were collected with 30° tilt angle. Data was collected at a rate of 120 movies per hour.

## Cryo-EM data processing and refinement

The POLG2 dataset was processed using cryoSPARC (*Punjani et al., 2017*) as detailed in *Figure 4—figure supplement 1C*. Final maps were sharpened using DeepEMhancer (*Sanchez-Garcia et al., 2021*). An atomic model from PDB ID: 2G4C (*Fan et al., 2006*) was fit into the map using Chimera (*Pettersen et al., 2004*).

## Data availability

Trained models used to obtain the results shown in *Figures 1 and 2* are available to download from 10.5281/zenodo.6842025. The Jupyter notebook used to aggregate the statistic in *Figure 3* and *Appendix 1—Tables 2–5* is part of the code repository (*Bouvette et al., 2022a*). Cryo-EM density maps of POLG2 collected using SmartScope have been deposited in the Electron Microscopy Data Bank (EMDB) with accession code EMD-25764. Square and hole images and corresponding labels used for training the ML models are available at 10.5281/zenodo.6814642 (square finder) and 10.5281/zenodo.6814652 (hole finder).

## Code availability

The source code for SmartScope is distributed under the open source BSD 3-clause license available at https://github.com/NIEHS/SmartScope (copy archived at swh:1:rev:9e58e2a2b278ca65156390175d-393819fbb16a3b, *Bouvette et al., 2022a*). The AI algorithms are available as a standalone package at https://gitlab.cs.duke.edu/bartesaghilab/smartscopeAI (copy archived at swh:1:rev:43b29ae8c333a-94463e0a4d9ecb97a5d5b6adf92; *Bouvette et al., 2022b*).

## Acknowledgements

This work was supported in part by the Intramural Research Program of the NIH; National Institute of Environmental Health Sciences (ZIC ES103326 and ZIA ES103341 to M.J.B., and Z01 ES065078 to W.C.C.), and a Visual Proteomics Imaging grant from the Chan Zuckerberg Initiative (2021-234602 to A.B.). This work utilized cloud computational resources and services accessed through the NIH STRIDES Initiative (https://cloud.nih.gov) and computational resources offered by Duke Research Computing (http://rc.duke.edu). We thank Tracy Futhey, Charley Kneifel, Katie Kilroy, Mike Newton, Victor Orlikowski, Tom Milledge, and David Lane from the Duke Office of Information Technology and Research Computing for assistance with the computing environment. We also thank David Fargo, John Grovenstein, and Chris Stone from the NIEHS Office of Scientific Computing for allocating computational resources to this project and especially to Gregory Stamper for assistance in setting up computing environments. We also thank Dr. Joshua Strauss from the UNC cryo-EM Core and Dr. Nilakshee Bhattacharya from the Duke Shared Material Instrumentation Facility (SMIF) for testing early versions of SmartScope. Molecular graphics and analyses performed with UCSF Chimera, developed by the Resource for Biocomputing, Visualization, and Informatics at the University of California, San Francisco, with support from NIH P41-GM103311. We thank Robin E Stanley and Bradley P Klemm for critical review of the manuscript.

## Additional information

### Funding

| Funder | Grant reference number | Author |
|---|---|---|
| National Institute of Environmental Health Sciences | ZIC ES103326 | Mario J Borgnia |
| National Institute of Environmental Health Sciences | ZIA ES103341 | Mario J Borgnia |
| National Institute of Environmental Health Sciences | Z01 ES065078 | William C Copeland |
| Chan Zuckerberg Initiative | 2021-234602 | Alberto Bartesaghi |

The funders had no role in study design, data collection and interpretation, or the decision to submit the work for publication.

### Author contributions

Jonathan Bouvette, Conceptualization, Software, Formal analysis, Investigation, Visualization, Methodology, Writing – original draft, Writing – review and editing, Development and implementation of SmartScope; Qinwen Huang, Software, Formal analysis, Investigation, Visualization, Methodology, Writing – original draft, Writing – review and editing, Developed and implemented the AI detection and classification for SmartScope; Amanda A Riccio, Formal analysis, Investigation, Visualization, Methodology, Writing – original draft, Writing – review and editing, PolG2 project including cloning and purification of the protein; William C Copeland, Resources, Supervision, Funding acquisition, Methodology, PolG2 project including conceptualization; Alberto Bartesaghi, Conceptualization, Resources, Supervision, Funding acquisition, Writing – original draft, Project administration, Writing – review and editing, Developed and implemented the AI detection and classification for SmartScope; Mario J Borgnia, Conceptualization, Resources, Supervision, Funding acquisition, Writing – original draft, Project administration, Writing – review and editing, Development and implementation of SmartScope

### Author ORCIDs

Jonathan Bouvette ⓘ http://orcid.org/0000-0003-3550-5319
Qinwen Huang ⓘ http://orcid.org/0000-0002-7082-5257
Amanda A Riccio ⓘ http://orcid.org/0000-0001-7782-0363
William C Copeland ⓘ http://orcid.org/0000-0002-0359-0953
Alberto Bartesaghi ⓘ http://orcid.org/0000-0002-7360-1523
Mario J Borgnia ⓘ http://orcid.org/0000-0001-9159-1413

### Decision letter and Author response

Decision letter https://doi.org/10.7554/eLife.80047.sa1
Author response https://doi.org/10.7554/eLife.80047.sa2

# Additional files

### Supplementary files

• MDAR checklist

### Data availability

SmartScope is open source software. Source code and installation instructions are available from https://github.com/NIEHS/SmartScope (copy archived at swh:1:rev:9e58e2a2b-278ca65156390175d393819fbb16a3b). Cryo-EM density maps of POLG2 collected using SmartScope have been deposited in the Electron Microscopy Data Bank (EMDB) with accession code EMD-25764.

The following datasets were generated:

| Author(s) | Year | Dataset title | Dataset URL | Database and Identifier |
|---|---|---|---|---|
| Bouvette J, Huang Q, Bartesaghi A, Borgnia MJ | 2022 | Automated systematic evaluation of cryo-EM specimens with SmartScope - Training data for square detector | https://doi.org/10.5281/zenodo.6814642 | Zenodo, 10.5281/zenodo.6814642 |
| Bouvette J, Huang Q, Bartesaghi A, Borgnia MJ | 2022 | Automated systematic evaluation of cryo-EM specimens with SmartScope - Training data for hole detector | https://doi.org/10.5281/zenodo.6814652 | Zenodo, 10.5281/zenodo.6814652 |
| Bouvette J, Huang Q, Bartesaghi A, Borgnia MJ | 2022 | Automated systematic evaluation of cryo-EM specimens with SmartScope - Trained models for square and hole detectors | https://doi.org/10.5281/zenodo.6842025 | Zenodo, 10.5281/zenodo.6842025 |
| Riccio AA, Bouvette J, Borgnia MJ, Copeland WC | 2022 | human DNA polymerase gamma accessory subunit, POLG2 | https://www.ebi.ac.uk/emdb/EMD-25764 | Electron Microscopy Data Bank, EMD-25764 |

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

## Appendix 1

## SmartScope: Framework for unsupervised cryo-EM imaging

**Appendix 1—table 1.** Input parameters for a SmartScope session.
Parameter names and their description. This is presented as a form on the web interface. These parameters can also be updated during the imaging process.

| Parameter name | Description |
| --- | --- |
| **General session parameters** | |
| Session name | Name of the microscopy session |
| Group | Group name of the microscopy session. Usually the principal investigator's name |
| Microscope | Microscope being used for the session |
| Detector | Detector being used for the session |
| **Collection parameters** | |
| Atlas X | Number of tiles for the Atlas acquisition in X axis (default: 3) |
| Atlas Y | Number of tiles for the Atlas acquisition in Y axis (default: 3) |
| Square X | Number of tiles for the window acquisition in X axis (default: 1) |
| Square Y | Number of tiles for the window acquisition in the Y axis (default: 1) |
| Squares num | Number of windows to be selected for high-magnification imaging (default: 3) |
| Holes per square | Number of targets per windows to be selected for higher-magnification imaging. If 0 is entered, all targets will be selected and data collection mode will be enabled (default: 3) |
| BIS max distance | Beam-Image shift grouping radius in microns (default: 0) |
| Min BIS group size | Smallest Beam-Image shift group size to be considered (default: 1) |
| Target defocus min | Lower end of the defocus range for rolling defocus in microns (default: –2) |
| Target defocus max | Higher end of the defocus range for rolling defocus in microns (default: –2) |
| Defocus step | Step by which the defocus is varied between each target group (default: 0) |
| Drift crit | Drift threshold to be met during the drift settling procedure before proceeding with high-magnification imaging. Use –1 to disable (default: –1) |
| Tilt angle | Tilt angle to use for high-magnification imaging. Works with BIS enabled (default: 0) |
| Save frames | Whether to save the movie frames or return aligned sum (default: Saving enabled) |
| Zeroloss delay | Time delay in hours for zero loss peak refinement. Only useful if the microscope has an energy filter. Use –1 to deactivate (default: –1) |
| Offset targeting | Enable random targeting off-center to sample the ice gradient and carbon mesh particles. Automatically disabled in data collection mode (default: enabled) |
| Offset distance | Override the random offset by an absolute value in microns. Can be used in data collection mode. Use –1 to disable (default: disabled) |
| **Autoloader (1 per grid)** | |
| Name | Name of the grid |
| Position | Position in the autoloader |
| Hole type | Grid hole spacing type (i.e. R1.2/1.3) |
| Mesh size | Grid mesh size and spacing (i.e. 300) |
| Mesh material | Grid mesh material (i.e. carbon or gold) |

**Appendix 1—table 2.** Examples of SerialEM settings for SmartScope usage.

These settings serve as guidelines and will need to be adapted for different hardware combinations. Updated versions of this table can be found at github.com/NIEHS/SmartScope.

| | Example 1 | Example 2 |
|---|---|---|
| **Instrument** | | |
| Microscope | Talos Arctica | Titan Krios G4 |
| Detector | Gatan K2 Summit | Gatan K3 |
| Energy filter | - | Gatan BioContiuum |
| **Low Dose Presets** | | |
| **Search** | | |
| Magnification | 210 | 580 |
| Pixel size (Å/pixel) | 196 | 152 |
| Mode | Linear | Counting |
| **View** | | |
| Magnification | 2600 | 8700 |
| Pixel size (Å/pixel) | 16.1 | 10.1 |
| Mode | Linear | Counting |
| **Focus/record** | | |
| Magnification* | 36,000 | 81,000 |
| Pixel size (Å/pixel)* | 1.19 | 1.08 |
| Mode | Counting | Counting |
| **Full grid montage presets** | | |
| Magnification | 62 | 135 |
| Pixel size (Å/pix) | 644 | 654 |
| Mode | Linear | Counting |

*These are the presets that are used for screening. They can be changes to suit the requirement for the specimen.

**Appendix 1—table 3.** Average precision obtained for each type of grid square.

| Feature type | Small | Cracked | Dry | Contaminated | Good | Partial |
|---|---|---|---|---|---|---|
| Without augmentation | 65.7% | 71.6% | 77.9% | 46.7% | 77.7% | 45.0% |
| With augmentation | 73.4% | 76.5% | 79.4% | 50.41% | 81.2 % | 45.5% |

**Appendix 1—table 4.** Smartscope screening mode statistics.

All grids were collected using a K2 detector. The default parameters for screening mode are a 9-tile atlas, 3 squares and 3 holes per square.

| Total specimens = 981 | min | max | mean | median | standard deviation |
|---|---|---|---|---|---|
| Squares sampled | 0 | 7 | 2.5 | 3.0 | 1.3 |
| Holes sampled | 0 | 33 | 8.0 | 9.0 | 5.0 |
| Time spent on specimen (min) | 3.0 | 56.3 | 21.0 | 20.9 | 8.4 |

**Appendix 1—table 5.** Smartscope screening mode statistics excluding specimens with no usable or visible squares.

All grids were collected using a K2 detector. The default parameters for screening mode are a 9-tile atlas, 3 squares and 3 holes per square.

| Total specimens = 772 | min | max | mean | median | standard deviation |
|---|---|---|---|---|---|
| Square sampled | 1 | 7 | 3.0 | 3.0 | 0.9 |
| Hole sampled | 1 | 33 | 9.4 | 9.0 | 4.0 |
| Time spent on specimen (min) | 9.1 | 56.3 | 23.0 | 21.7 | 6.7 |

**Appendix 1—table 6.** Smartscope data collection mode statistics.

Start times are at the start of specimen loading in the column. All grids were imaged using a K2 detector.

| Total specimens = 58 | min | max | median | mean ± SD |
|---|---|---|---|---|
| Total micrographs | 565 | 8333 | 1626 | 2155±1502 |
| Holes per hour | 51 | 141 | 100 | 100±22 |
| Data collection setup time (min)* | 6 | 80 | 32 | 34±17 |
| Setup time per 1000 micrographs (min) | 4 | 55 | 17 | 19±11 |

*Data collection times are calculated as the time needed from grid loading to the collection of the first 50 high-magnification targets minus the time required for these 50 targets to be acquired.

