## [Editor Report]

This paper describes a new software tool: SmartScope, for automated screening of cryo-EM grids. SmartScope can also perform automated data collection on suitable grids, including with beam-image shifts and tilted stage geometries. If it works in practice as advertised in the paper, then it will be a highly useful tool for the field, especially if other groups would also contribute to its open-source and modular code.

---

## [Decision Letter]

**Decision letter after peer review:**

Thank you for submitting your article "Automated systematic evaluation of cryo-EM specimens with SmartScope" for consideration by *eLife*. Your article has been reviewed by 3 peer reviewers, including Sjors HW Scheres as the Reviewing Editor and Reviewer #1, and the evaluation has been overseen by a Reviewing Editor and Kenton Swartz as the Senior Editor. The following individual involved in the review of your submission has agreed to reveal their identity: Arjen J Jakobi (Reviewer #3).

Essential revisions:

All three reviewers agreed that the software described in this paper would be a highly useful addition to the field and are in support of publication.

1) A table summarising the data acquisition parameters and data collection statistics should be provided.

Strong encouragements:

1) The reviewers praise the authors for their intention to make the code available to the community and encourage them to do so as soon as possible.

2) Besides providing the source of the smartScope program on github, the authors are also encouraged to provide the labelled raw training data for the convolutional neural networks. Public repositories like Zenodo or EMPIAR could be used.

3) The 3 reviewers each make useful suggestions to improve the manuscript below. Although these are not essential for acceptance, the authors are encouraged to give this serious consideration.

*Reviewer #1 (Recommendations for the authors):*

All Supplementary Figures should become supplements to one of the main figures.

*Reviewer #2 (Recommendations for the authors):*

The paper is in my opinion very well structured with clear figures and procedure steps. In general, I really enjoyed reading it and I am willing to test it soon. Of course, a work of this type passes the real test when it starts being used by the community. I believe that SmartScope will be used as there are no other fully automated solutions for grid screening.

From the description in the text, the software appears to be quite solid and the authors have pretty much answered in the text itself any questions that were coming to my mind while reading. In short, this manuscript is very detailed, easy to follow in a logical mind flow, and takes into consideration all aspects of specimen screening and data acquisition. From my side, the article is publishable as it is.

Some more detailed comments:

Line 143: SmartFlow is bound to serialEM, will it be possible to couple it to EPU for example, which is widely used, or other software used for the collection of diffraction data? How dependent is on serialEM 4.0? Will SmartScope be maintained as serialEM develops?

Line 153: Is there a minimal magnification that has to be used for the window clustering to work properly? Can the authors include some comments/guidelines on this?

Line 175: Do I understand correctly that some images are also taken in the carbon, or at the carbon/ice interface to evaluate the distribution of particles?

Line 179: I would simplify the procedure to gain time, at least for relatively short exposure times. Probably not in all cases frames alignment and CTF estimation are necessary, to view the stability of the new microscopes equipped with autoloader. However, it is probably better to have these options while running the automatic screening e.g. overnight.

Line 221: Out of full completeness the authors might comment on how their algorithm to maximise target coverage while minimising numbers of groups compares to the same procedure in EPU. There the BIS applied can be changed. Is that possible in SmartScope?

The fact that SmartScope can perform BIS collection on tilted specimens in a correct way is great.

For the evaluation of the network organisation part, I have asked the opinion of an expert colleague, who commented:

Line 237: I don't quite understand what the authors mean with: "bundles a webserver and the main imaging workflow". I assume the webserver is for interacting with the program and the "main imaging workflow" is what runs on the worker if installed on separate computers, but it would be better to make it a bit more clear.

The Singularity container image is convenient and good that it's versatile enough to be installed on separate systems.

Line 245: Why do both the web server and the worker need to have access to a shared disk? Does the worker write and the web server only reads the results to display to the user? What kind of information is stored in the database?

248: The object store can be accessed via the Amazon S3 API. It's a nice feature for exporting data to "the cloud". Maybe the author should clarify a bit what are the full advantages that one can get from such a setup?

Supplementary 43: "communicate with each other using Socket or SSH connections" Do they mean HTTP? In the graph, there is no mention of SSH connections, only HTTP.

*Reviewer #3 (Recommendations for the authors):*

Points for consideration:

– The introduction describes many important concepts and limitations in the preparation of cryo-EM specimens and in screening/data collection, but only sparsely cites, and if, very general reviews on these topics. The reader may benefit if the authors would more specifically refer to the excellent primary literature on these topics.

– In addition to their YOLO-based hole finder, the authors may consider mentioning that the open and modular approach of their software allows integration with approaches such as e.g. virtual maps in Py-EM, which could extend the reach of their method to more sophisticated screening/targeting scenarios and different sample supports.

– The micrograph pre-processing routines implemented in SmartScope currently involve frame alignment and CTF estimation. The authors may consider including the possibility of image denoising workflows such as those implemented in data pre-processing pipelines (Warp, Scipion, SPHIRE, CryoSparc Live, …), which can be useful tools for rapid selection of suitable imaging areas, in particular if particles are small.

– The authors comment on the screening mode statistics from the operation of SmartScope in their facility. From their facility projects, are there statistics available on how many cases the screening procedure followed by automated, targeted data collection lead to successful structure determination, and how this compares to manual screening and subjective targeting on the same sample? This could be insightful.

– The authors mention that data collection of human mitochondrial DNA polymerase involved tilting of the specimen in a subset of the dataset to improve angular sampling. It would be useful to show angular distribution plots for the data excluding and including the tilted images to illustrate this was required and how automation by SmartScope can help detect and mitigate such problems on the fly.

– A table summarising the data acquisition parameters and data collection statistics should be provided.

– It is laudable that the authors make their software and models publicly available. It would be more useful to have the repository public open at the moment of preprint posting so as to give reviewers the possibility to also screen review code. The data availability statement contains a remark of disclosure of data upon reasonable request. It is unclear what this statement means, and which requests are considered reasonable; these data can be useful for other academic projects following related but complementary approaches, as well as comparison and benchmarking. I encourage the authors to make raw ML data available through public repositories such as Zenodo, and to deposit raw micrographs at EMPIAR.

---

## [Author Response]

Essential revisions:All three reviewers agreed that the software described in this paper would be a highly useful addition to the field and are in support of publication.

We appreciate the support of the reviewers.

1) A table summarising the data acquisition parameters and data collection statistics should be provided.

Thank you for pointing this out, we have added this information in the new table1.

Strong encouragements:1) The reviewers praise the authors for their intention to make the code available to the community and encourage them to do so as soon as possible.

Thank you. The code will be made available immediately when the manuscript becomes live. The Code availability section now states that the code will be available and the corresponding github URL.

2) Besides providing the source of the smartScope program on github, the authors are also encouraged to provide the labelled raw training data for the convolutional neural networks. Public repositories like Zenodo or EMPIAR could be used.

Thanks for pointing this out, this was our intention from the beginning, which we considered implicit in committing to the publication of the code. The raw training data is now available on Zenodo. The data availability section of the manuscript has been modified to refer explicitly to the training datasets as well as to the trained model.

3) The 3 reviewers each make useful suggestions to improve the manuscript below. Although these are not essential for acceptance, the authors are encouraged to give this serious consideration.

We appreciate the suggestions and we have introduced changes guided by them.

Reviewer #1 (Recommendations for the authors):All Supplementary Figures should become supplements to one of the main figures.

We have reorganized the figures as requested.

Reviewer #2 (Recommendations for the authors):The paper is in my opinion very well structured with clear figures and procedure steps. In general, I really enjoyed reading it and I am willing to test it soon. Of course, a work of this type passes the real test when it starts being used by the community. I believe that SmartScope will be used as there are no other fully automated solutions for grid screening.From the description in the text, the software appears to be quite solid and the authors have pretty much answered in the text itself any questions that were coming to my mind while reading. In short, this manuscript is very detailed, easy to follow in a logical mind flow, and takes into consideration all aspects of specimen screening and data acquisition. From my side, the article is publishable as it is.Some more detailed comments:Line 143: SmartFlow is bound to serialEM, will it be possible to couple it to EPU for example, which is widely used, or other software used for the collection of diffraction data? How dependent is on serialEM 4.0? Will SmartScope be maintained as serialEM develops?

Historically, SerialEM didn’t remove features so we expect that all the current scripting commands within SerialEM4.0 will exist for the foreseeable future. We are planning on making SmartScope a long-term solution where more features and updates will be added.

Line 153: Is there a minimal magnification that has to be used for the window clustering to work properly? Can the authors include some comments/guidelines on this?

We added a supplementary table 2 with the magnification/pixel sizes that have been used for acquiring the squares. The hole finder was also successfully tested on the low-mag atlas but is not part of the software yet.

Line 175: Do I understand correctly that some images are also taken in the carbon, or at the carbon/ice interface to evaluate the distribution of particles?

Intermediate magnification images always include carbon areas (for carbon grids naturally). The version that we tested for this version of the manuscript did not contemplate taking images of the interface at higher magnification. In a newer version, now available, random exploration of the hole area including the water carbon interface can be achieved by specifying an “offset” parameter from the center of the hole, we have amended the manuscript to reference this parameter. Future versions may include “smarter” exploration based on the analysis of intermediate resolution pictures.

Line 179: I would simplify the procedure to gain time, at least for relatively short exposure times. Probably not in all cases frames alignment and CTF estimation are necessary, to view the stability of the new microscopes equipped with autoloader. However, it is probably better to have these options while running the automatic screening e.g. overnight.

Frame alignment is not essential and will occur only when the exposure mode is set in SerialEM to save the frames. However, since these processes occur asynchronously from acquisition, movie alignment does not have an impact in the speed of imaging.

Line 221: Out of full completeness the authors might comment on how their algorithm to maximise target coverage while minimising numbers of groups compares to the same procedure in EPU. There the BIS applied can be changed. Is that possible in SmartScope?

SmartScope will group the targets according to an input BIS radius in the acquisition settings, which can be changed at any point during the session. A sentence was added in the manuscript to clarify this point. Interestingly, newer versions of EPU seem to have eliminated the user’s ability to modify this parameter in the GUI. The BIS (AFIS) radius in EPU can still be changed through parameter hidden in the global configuration files, but requires restarting the software. The comparison with the algorithm used in EPU is difficult as the code for EPU is not available.

The fact that SmartScope can perform BIS collection on tilted specimens in a correct way is great.For the evaluation of the network organisation part, I have asked the opinion of an expert colleague, who commented:Line 237: I don't quite understand what the authors mean with: "bundles a webserver and the main imaging workflow". I assume the webserver is for interacting with the program and the "main imaging workflow" is what runs on the worker if installed on separate computers, but it would be better to make it a bit more clear.

We have reworked this section to improve clarity.

The Singularity container image is convenient and good that it's versatile enough to be installed on separate systems.Line 245: Why do both the web server and the worker need to have access to a shared disk? Does the worker write and the web server only reads the results to display to the user? What kind of information is stored in the database?

Only metadata about the relationship between the images and their targets is stored in the database. All the images are served directly from disk. This helps in keeping the size of the database small.

248: The object store can be accessed via the Amazon S3 API. It's a nice feature for exporting data to "the cloud". Maybe the author should clarify a bit what are the full advantages that one can get from such a setup?

This feature allows to set up mirror instance in the cloud to provide access to users who, for security reasons, may not have access to the resources behind a firewall. At this point, this instance is distinct from the system that allows navigation of the specimen in real time.

Supplementary 43: "communicate with each other using Socket or SSH connections" Do they mean HTTP? In the graph, there is no mention of SSH connections, only HTTP.

SSH has been added to the figure to match the description.

Reviewer #3 (Recommendations for the authors):Points for consideration:– The introduction describes many important concepts and limitations in the preparation of cryo-EM specimens and in screening/data collection, but only sparsely cites, and if, very general reviews on these topics. The reader may benefit if the authors would more specifically refer to the excellent primary literature on these topics.

Heeding the request of Reviewer #1, we have shortened significantly the introduction removing most of the discussion on the difficulties of specimen preparation. Instead, we have introduced a couple of key references.

– In addition to their YOLO-based hole finder, the authors may consider mentioning that the open and modular approach of their software allows integration with approaches such as e.g. virtual maps in Py-EM, which could extend the reach of their method to more sophisticated screening/targeting scenarios and different sample supports.

We have added a citation to Py-EM to the paragraph in which we discuss modularity.

– The micrograph pre-processing routines implemented in SmartScope currently involve frame alignment and CTF estimation. The authors may consider including the possibility of image denoising workflows such as those implemented in data pre-processing pipelines (Warp, Scipion, SPHIRE, CryoSparc Live, …), which can be useful tools for rapid selection of suitable imaging areas, in particular if particles are small.

This is indeed a feature that we’re interested in adding to replace the current preprocessing when necessary. All the pieces are in place to build different wrappers around other packages to fetch the processing data. We didn’t focus on these yet because they didn’t seem quite as urgent for processing a few screening images. However, it becomes more interesting when thinking about data collection.

– The authors comment on the screening mode statistics from the operation of SmartScope in their facility. From their facility projects, are there statistics available on how many cases the screening procedure followed by automated, targeted data collection lead to successful structure determination, and how this compares to manual screening and subjective targeting on the same sample? This could be insightful.

We did not run a control comparison between SmartScope and manual operation. One of the main reasons is that, in the case of data collection, humans are still the ones deciding which are the squares to collect.

– The authors mention that data collection of human mitochondrial DNA polymerase involved tilting of the specimen in a subset of the dataset to improve angular sampling. It would be useful to show angular distribution plots for the data excluding and including the tilted images to illustrate this was required and how automation by SmartScope can help detect and mitigate such problems on the fly.

The intent was not to provide a solution to a problem with this particular sample but rather demonstrate that the quality of tilted data approximates that of untitled images. As it turns out, tilting was not required for POLG2. It was our first time collecting the samples so a subset of the data was collected with tilt to ensure a better angular distribution. We added a breakdown of the tilted vs non-tilted resolutions and angular distribution in the table 1 and Figure 1—figure supplement 2 The particles from the tilted data alone reached a 3.9A resolution and although their inclusion did not improve the to the overall final resolution in the combined dataset, the distribution plot indicates improvements in angular sampling.

– A table summarising the data acquisition parameters and data collection statistics should be provided.

This information was added in the new table 1.

– It is laudable that the authors make their software and models publicly available. It would be more useful to have the repository public open at the moment of preprint posting so as to give reviewers the possibility to also screen review code. The data availability statement contains a remark of disclosure of data upon reasonable request. It is unclear what this statement means, and which requests are considered reasonable; these data can be useful for other academic projects following related but complementary approaches, as well as comparison and benchmarking. I encourage the authors to make raw ML data available through public repositories such as Zenodo, and to deposit raw micrographs at EMPIAR.

We decided to go through a closed beta phase to address questions of ease of installation, documentation and testing at other facilities. Adoption of a new software relies on making sure these points are addressed before release. The code will be made available via github upon publication, and the training data and trained models will be made available on Zenodo. We have made changes to the data and code availability sections to reflect these changes and reported the datasets in the manuscript submission system. We agree with the reviewer that it would be a good feature for the journal to provide means of ensuring access to private repositories by anonymous reviewers.